# The Effect of Comfort- and Hot-Period on the Blood Flow of Corpus Luteum (CL) in Cows Treated by an OvSynch Protocol

**DOI:** 10.3390/ani11082272

**Published:** 2021-07-31

**Authors:** Isfendiyar Darbaz, Serkan Sayiner, Osman Ergene, Kamil Seyrek Intas, Feride Zabitler, Enver Cemre Evci, Selim Aslan

**Affiliations:** 1Department of Obstetrics and Gynecology, Faculty of Veterinary Medicine, Near East University, Nicosia 99138, North Cyprus, Turkey; osman.ergene@neu.edu.tr (O.E.); kamil.seyrekintas@neu.edu.tr (K.S.I.); feride.zabitler@neu.edu.tr (F.Z.); envercemre.evci@neu.edu.tr (E.C.E.); selim.aslan@neu.edu.tr (S.A.); 2Department of Biochemistry, Faculty of Veterinary Medicine, Near East University, Nicosia 99138, North Cyprus, Turkey; serkan.sayiner@neu.edu.tr

**Keywords:** dairy cow, color Doppler, heat stress, corpus luteum, luteal blood flow, IGF-1, progesterone

## Abstract

**Simple Summary:**

Doppler ultrasonography is frequently used to measure blood flow. The Ovsynch program is applied to synchronize the timing of ovulation in dairy cows. Heat stress can negatively affect the hormonal balance, ovarian activity, and blood flow. In this study, the effect of heat stress on corpus luteum blood flow, progesterone, and insulin-like growth factor parameters was investigated during and after Ovsynch synchronization. Our results showed that synchronization initiated with high progesterone values caused significantly higher blood flow and greater corpus luteum area in the comfort period when compared with the hot period. In addition, insulin-like growth factor values were found significantly higher during the comfort period compared to heat stress. Under heat stress circumstances, the Ovsynch synchronization provided better results when the progesterone levels were high. We suggest that it may be better to apply the modified Ovsynch program to increase progesterone levels in cows with low progesterone values when the protocol is initiated during the heat stress period.

**Abstract:**

The values of luteal blood flow (LBF), total corpus luteum (CL) area (TAR), and progesterone (P4), during and after OvSynch (OvS) protocol in comfort (CP; *n* = 40) and hot periods (HP; *n* = 40) were compared. We investigated how low and high P4 values obtained before the application affected the parameters above during CP and HP periods. Blood samples were collected before the OvS application on day 0 (OVSd0), day 9 (OeG), and day 18 (9th day after OeG: OvSd9). The P4 (ng/mL) values of the animals exhibiting dominant follicles were between 0.12–0.82 in HC and 0.1–0.88 in CP (P4-2: 4.36–4.38 and P4-3: ≥7.36 ng/mL). The LBF values were measured on days 7 (OvSd7) and 9 (OvSd9) after the OeG. The P4 mean values at day 0 (OvSd0) were classified as low (P4-1), medium (P4-2), and high (P4-3). The LBF and the TAR values in the P4-2 and P4-3 on OeG day 9 were higher than in HP (*p* < 0.05; 0.001), but there was no significant difference in the P4-1. In conclusion, when the OvS program was initiated with low P4 values, no difference was observed between HP and CP in terms of LBF values; however, when the program was started with high P4 values, there were significant increases in LBF and TAR values in the CP compared to the HP.

## 1. Introduction

Heat stress negatively affects fertility as temperatures rise [1]. When comfort and hot periods are compared, the pregnancy rate during the first insemination was 44% and 27%, and the anestrus rate was 1.2% and 12.9%, respectively, with results favouring the cooler period [2].

Summer heat stress lowers the fertility in cattle in hot environments by influencing oocyte quality, follicular activity, and blood plasma progesterone (P4) levels [1,3]. However, the mechanisms by which elevated temperatures influence the corpus luteum (CL) functioning remain unclear. The upper limit of ambient temperature at which lactating dairy cows can maintain a stable body temperature (upper critical temperature) is as low as 25–27 °C. Therefore, heat stress is not confined to tropical regions of the world and imposes a considerable cost on milk and beef production [3].

It has been revealed that the comfort zone for cows is between 5 and 25 °C, and the general condition and productivity of animals is not negatively affected at temperatures in this range [4,5]. Programs that stimulate ovulation by coordinating the development of the follicle and CL have been successfully implemented in lactating cows. Timed insemination programs, which eliminate the difficulties in estrus detection, enable the cow to be inseminated in estrus at a specified time and induce pregnancy [6]. Silent estrus is a severe problem in large herds during periods of heat stress [7]; previous research suggests that this program or its modified version is also successful during periods of heat stress, leading to increased pregnancy rates [8,9]. Although the OvSynch (OvS) program increases fertility performance during heat stress, it cannot prevent embryonic deaths [8].

Color Doppler Ultrasonography (CDU) has long been used to measure the physiological changes and functions of the ovary and uterus [7]. For this purpose, with this technique, measurements of real-time changes in luteal blood flow (LBF) during induction [7,10] or spontaneous [11] luteolysis are made in cows. CDU has also been applied in cows to investigate the relationship between LBF, luteal size (LS), and P4 [7].

In this study, the effect of heat stress in comfort and hot periods was investigated in dairy cows, one of the most significant breeding animals for human health and national economies. In the light of data from previous studies, the aims of the present study were i) to measure LBF, TAR, and the blood P4 values after OvS protocol in comfort and the hot periods and ii) to assess differences of these parameters exploring new approaches in the management of heat stress, revealing how especially low and high progesterone values affect the OVS protocols in comfort and hot periods.

## 2. Materials and Methods

The local animal ethics committee of Near East University approved the study (Decision No: 7/24.10.2016), and the ethical guidelines mentioned in the 1964 Declaration of Helsinki and its later amendments were followed. The study was carried out on a commercial dairy farm located in Ercan, Nicosia, North Cyprus, following the approval of the Veterinary Department of the Ministry of Agriculture and Forestry, North Cyprus.

### 2.1. Animals

The study was performed during the following two periods; Comfort period (CP, winter (from December 2018 to February 2019), average temperature, precipitation, and humidity for 3 months: respectively 11.5 °C (min 2.5–max 19.7 °C), 142 mm, 74.0%; *n* = 40), and Hot period (HP, summer (from June 2018 to August 2018), average temperature, precipitation, and humidity for 3 months: respectively 35.5 °C (min 28.2–max 40.1 °C), 3.4 mm, 64.0%; *n* = 40). Additionally, there are no cooling or ventilation systems present on the farm. A total of 80 Holstein Friesian cows were assigned to this study (body condition score 2.75 and 3.00). The cows were kept at the same farm and provided with similar housing conditions and feeding regimes to exclude the influence of confounding factors such as feeding and environment. The average age of the study animals was 5 ± 1.5 years (3–8); all were multiparous, with an average milk yield of 27.0 ± 3.7 (min 22.0–max 35.0) litres. The animals were housed in a free-stall system and had ad libitum access to feed and water. The diet consisted of a base ration given ad libitum as a daily total mixed ration (TMR). The feeding frequency of TMR was twice per day. During the dry period, the ration was based on hay, green grass, TMR, an admixture of minerals and vitamins, and it was adjusted to contain 10–13% crude protein, 2–3.5% crude fat, and 35–40% neutral detergent fiber on a dry matter (DM) basis. The post-partum diet contained corn silage, hay, green grass, TMR, an admixture of minerals and vitamins, adjusted to contain 17–20% crude protein, 3–5% crude fat, and 27–34% neutral detergent fibre on a dry matter (DM) basis. 

Animals were screened before starting the applications (OvSd0). General (BCS 2.75–3.0; fat/protein ratio 1.16–1.18) and gynecological controls were performed (uterus symmetrical/close to symmetrical, no thickening, active ovaries, active CL/follicles, no vaginal discharge) of the animals. The status obtained from the control of animals is given in Table 1. It was determined that there was no difference between the parameters such as lactation number, days in milk, milk yield, and body condition score between the HP and CP groups. 

### 2.2. Experimental Design 

The OvS protocol [12] was applied to start from the 35th day post-partum. LS, LBF, and P4 values were measured in an OvS protocol. The blood samples were collected (Vena jugularis) into serum separator tubes (SST) from each cow on day 0 (before the start of the application: OvSd0), day 9 (estrus: OeG), and day 18 (9th day after OeG: OvSd9). The first GnRH application was made after the blood was collected on day 0 (OvSd0) (Table 2). Sera were obtained following centrifugation at 1500 g for 10 min at +4 °C and stored at −80 °C until the analysis. The classification of P4 values was done retrospectively as low, medium, and high after P4 analyses were performed in the laboratory at the end of the study. The P4 (ng/mL) mean values at day 0 (OvSd0) were classified as follows: HP: 0.64 (0.10–1.42; *n* = 19), CP: 0.85 (0.16–1.97; *n* = 18) = (P4-1); HP: 4.36 (2.18–5.62; *n* = 12), CP:4.38 (2.56–5.80; *n* = 10), (P4-2); and HP: 7.36 (6.23–8.59; *n* = 9), CP: 10.33 (6.79–14.30; *n* = 12), (P4-3). 

The animals were not inseminated during estrus after the last GnRH administration (OeG). CL development on the 9th day postestrus was examined, and the LBF values were measured on days 7 (OvSd7) before PGF2-α administration (D16) and 9 (OvSd9) after the estrus (D18) (Table 2). Estrus (OeG) was determined according to known estrus criteria in follicle measurements (1.4–2.23 cm) and uterus examination, which was performed 24 h after the last GnRH administration [13,14]. 

### 2.3. Measurement of Luteal Blood Flow (LBF) and Total CL Area (TAR)

LBF of the CL was measured using a Color Doppler (portable LOGIQ Book XP ultrasound device-General Electrics Healthcare, Solingen, Germany; 10 MHz linear probe- Model 1739RS, Tokyo, Japan). Ultrasonographic controls and pictures taken were processed as described previously [14,15]. Pictures were frozen at the maximum cross-section and saved for measurements. The pictures were then examined using the same instrument equipped with a linear probe for imaging BF of CL in power Doppler mode (Doppler frequency: 5 MHz; gain: 19.5; pulse repetition frequency: 0.5 kHz). Five pictures without flash artefacts and a maximum number of colored areas were stored in system memory in the DICOM (Digital Imaging and Communications in Medicine) format. The stored Doppler images (5 images of blood flow regions in the LC) were analyzed using a specialised software program (Pixel Flux, version 1.0, Chameleon Software, Leipzig, Germany). For this, the entire CL structure and blood flow region was selected as the regions of interest and the colored areas were calculated (Figure 1). An average of five images was used to evaluate LBF further. 

### 2.4. Progesterone (P4) and Insulin-Like Growth Factor-1 (IGF-1) ELISA Analyses

P4 and IGF-1 concentrations were measured using enzyme immunoassay test kits (Progesteron, DE1561, Lot. 23K039-2, Demeditec, Kiel, Germany; RayBio^®®^ Bovine IGF-1 ELISA, ELB-IGF-1, Lot 120117 0659, Raybiotech, Inc., Peachtree Corners, GA 30092, USA). The tests were carried out following the manufacturer’s directions. The washing steps of the ELISA test were performed using an automated microtiter washer (MW-12A Microplate washer, Mindray, Shenzhen, China), and the results were obtained using a microtiter plate reader at 450 nm (MR-96A Microplate reader, Mindray, Shenzhen, China). The intra- and inter-assay coefficients of variation (CV) were 2.9% and 4.0% for P4, and 5.8% and 8.5% for IGF-1, respectively. 

### 2.5. Statistical Analyses 

In the statistical evaluation, all analyses were carried out using the IBM SPSS software (version 21). The mean value and the standard deviations were obtained by applying descriptive statistics (X ± SD). The Shapiro–Wilk test was used to determine the homogeneity distribution. The Mann–Whitney U test was used as the nonparametric test in the case of non-homogeneous distribution of the values. In the case of homogeneous distribution, the Independent Samples T-test was used. A Chi-square test was used in the proportional comparisons. General Linear Models-Repeated Measures-Test was used for the independent values. Bivariate analysis was applied as correlation, and the Pearson test was selected as the Correlation Coefficient. A *p*-value of <0.05 was considered statistically significant.

## 3. Results

There was no difference (*p* > 0.05) between the LBF-HP and the LBF-CP values on the 7th and 9th days after the OvS protocol (Table 3).

On the 9th day after OvS, there was no statistically significant difference between the HP and the CP periods in terms of both P4-Serum and IGF1-Serum values (*p* > 0.05; Table 4).

The TAR measurements showed a significant difference at the level of *p* < 0.0001 between the CP and the HP values both on the seventh and the ninth days after OvS. It was found that the CP values on both days were greater than the HP values (Table 5).

No statistical difference was found in the follicle diameters between the HP and the CP. On the other hand, the IGF-1 values in the comfort period were significantly higher than those from the summer period (Table 6).

As a result of the examination of blood samples obtained 24 h after the last GnRH administration (OeG) in the OvS protocol, it was determined that the IGF-1 values in the summer period were significantly lower than in the CP (*p* < 0.01). However, there was no significant difference in follicle size between the two periods. 

According to the classification of the initial P4 values at the beginning of the OvS protocol (P4-1, P4-2, and P4-3), when the LBF values corresponding to these values were examined on the 9th day after OVS, it was determined that the LBF values that belonged to the P4-2 and P4-3 categories were significantly higher in the CP period than the HP period. In addition, it was determined that the CL size corresponding to the initial P4-1, P4-2, and P4-3 values 9 days after OvS were significantly higher in the CP period than the HP period (*p* < 0.05; 0.001 and 0.0001; Table 7).

While there was no significant correlation between the P4 grouping, based on samples measurements at the start of the OvS protocol, and LBF and TAR values corresponding to that grouping in the HP group, a significant positive correlation was determined between the P4 categories (P4-1, P4-2, P4-3), LBF, and TAR (0.999; *p* < 0.05 and *p* < 0.01) in the CP period, and a high level of correlation between LBF and P4, again in the CP period (1.0; *p* < 0.001), was detected (Table 8).

## 4. Discussion

The negative effect of heat stress on both the animal and its reproductive system has long been known [16,17,18]. In studies on the effect of heat stress on steroid hormones, while it has been determined that plasma estrogen concentrations decrease due to heat stress [1,19,20,21], studies on blood progesterone levels are contentious. While some studies suggest no change as a result of heat stress [21], other studies indicate both decreases [22,23,24] or increases [25,26]. 

Various synchronization programs are applied to create timed artificial insemination. Among these programs, the OvSynch (OvS) protocol has been used successfully in synchronizing ovulation in timed insemination of cows during the first service [6,13]. Since this study aimed to measure the LBF values and TAR in the luteal stage, the animals were not inseminated. 

In our study, the data obtained as a result of OvSynch synchronization were evaluated in two ways: During the OvS protocol and in the luteal periods after completion of the application, as the calculation of the differences between HP and CP without considering the P4 values at the beginning of the application, and as considering the P4 values in categories as low, medium, and high (P4-1, P4-2, P4-3) at the beginning of the application. Some research suggests that the OvS protocol increases pregnancy rates when applied during the summer [9,27]; however, other research indicates better results are obtained in winter as opposed to summer [28]. Based on these reasons, we investigated the differences obtained in terms of LBF and TAR in OvS applications based on P4 values in HP and CP periods. 

In our study, when the OvS protocol was initiated, there was no statistically significant difference between the HP and CP in terms of P4 values. No statistically significant difference was found between the HP and CP in terms of serum progesterone measurements performed on the 9th day after the end of the OvS protocol. This result supports previous publications that stated that heat stress has no effect on P4 values [29,30]. It has been reported that conditions such as hepatic metabolism, other stress conditions, and dry matter intake have a greater effect on progesterone levels [1].

Data obtained as a result of Color Doppler Ultrasonography LBF measurements applied during the CL formation found no statistically significant difference between HP and CP in terms of the LBF and the P4 values. It has been demonstrated that heat stress causes a 20–30% reduction in blood flow in the ovaries of rabbits and poultry [31,32]. Conversely, there are publications related to cows that have found heat stress does not affect the reaction of the CL and P4 circulation [33] and that rising temperatures do not negatively affect the luteal function and luteal cells [3] Previous research has revealed that P4 values are lower in the summer during the luteal period of the estrus cycle [23], or P4 values are higher in winter on the 9th day of in vitro luteinization [34]. In our study, when the results obtained in the luteal period after OvS protocol were calculated (regardless of the difference between P4 concentrations at the beginning of the application), it was concluded that the LBF and the P4 values were not affected by heat stress. To our knowledge, there are no publications to date in which the differences between the LBF or the CL sizes according to the season were examined with color Doppler data. Only one publication has shown less dominant follicular blood flow in cows with heat stress than in cows under cold management [35].

Although LBF and P4 were not affected, it was revealed that the TAR was higher during the CP period, both on the 7th and the 9th days after the OvS protocol. No significant relationship was determined between the CL and the P4 values in the early and mid-luteal periods. A significant association could be determined 5 days before ovulation [36]. The maximum number of colored pixels in the cross-sectional area of the CL revealed a correlation between progesterone and CL change during the cycle process, but it was also revealed that the CL size did not reflect the blood flow and peripheral blood P4 alteration [37]. In our study, the size of CL did not increase between the 8th and the 16th days of the cycle; however, there was an increase in the progesterone concentration between these days. 

There is a decrease in blood flow shortly after ovulation, but with angiogenesis on the 2nd–5th days of the cycle, progesterone also increases with an increase in CL volume [10]. However, in this study, the CL measurements were made on the 7th and 9th days after the OvS protocol. Many factors support the function of CL. Luteinizing Hormone (LH), and Growth Hormone (GH) are necessary for the development and function of CL. In addition, angiogenic factors such as vascular endothelial growth factor (VEGF) and fibroblast growth factor (FGF) are also effective [38]. 

In our study, no statistically significant difference in follicle diameter was found between CP (1.67 cm) and HP (1.52 cm). However, IGF-1 measurements made on the same day showed that the CP values were significantly higher than the HP measurements during this period (*p* < 0.01). Various studies have found no relationship between the follicle diameter and IGF-1 [39,40]. It has been determined that heat stress negatively affects mostly the follicular fluid, the oocyte, and the granulosa cells [41]. Heat stress affects follicle selection and extends the follicle wave duration, resulting in a decrease in the quality of oocytes [42]. These results show that follicle content and structure are affected rather than the follicle size, which supports our findings.

IGF-1 values are lower in the summer than in the thermoneutral period [43,44]. IGF-1 plays a role in developing follicles in synergy with follicle-stimulating hormone (FSH) and LH and supports steroidogenesis by enhancing the LH receptor induction and inhibin synthesis [45]. IGF-1 also increases the sensitivity of follicular cells to FSH and LH [46]. This is important in follicle development and improvement of oocyte quality [47]. In our findings, IGF-1 values in the CP are significantly higher than in the HP in blood measurements made during the follicle period 24 h after the last GnRH administration.

However, in this study, cows with high P4 values (P4-2 and P4-3 values) revealed that the average blood flow in CL on the 9th day post-application was significantly higher in the CP period. When the P4 value is high at the beginning of OvS (no difference between CP and HP P4 values during start treatment), which develops 9 days after the last application, there is a different response in the LBF between the two periods with an increase during the CP period.

As P4 values increased, there was a significant increase in both LBF (r = 0.999) and TAR (r = 0.999) values during the CP period (*p* < 0.05), while this increase did not occur during the HP period. An interesting result from the study is that the TAR values were significantly higher in the CP than HP according to the P4 categories. In some publications, there is a correlation in the measurements made between P4, LBF, and TAR [14,15]. The detection of a correlation in this study during the CP period, but the absence of such a correlation during the HP period indicates that this interaction occurs during the appropriate environmental conditions (CP). It seems that when OvS protocol is started with high progesterone values, high BF and larger CL sizes can be obtained in CP. The OvS protocol can always be started, but when started in the early diestrus period (days 5–12), it has been shown that the ovulation rate is 70% higher than during other periods (53%) [48]. It has been shown that when the OvS protocol is initiated on days 5–12, the rates of conception also increase [49,50,51]. In another study [52], in animals without CL, after waiting for 7 days, the pregnancy rate was found to be 40.9% in the OvS protocol, which was started after the CL was determined, while the pregnancy rate in animals without CL was found to be 25.6%. In our findings, as a result of OvS protocol performed in two categories (P4-2 and P4-3) where progesterone was high, significant increases in LBF and TAR values during CP also support these findings. In addition, studies demonstrate that the data obtained from the OvSynch protocol are better when performed during the winter period compared to those performed in the summer period [53]. While the conception rate of the first artificial insemination (AI) was 17.5% in heat stress (HS) from the OvS protocol, this rate was 23.3% in the winter season (WS). Likewise, while days open were 179.6 days in HS conditions, it was found as 104.02 days (*p* < 0.05) in WS (52). One study showed the conception rate obtained from the OvS protocol in the HS period was found to be significantly (*p* < 0.05) lower (25% vs. 35%) compared to the WS period [28]. When the OvSynch protocol was administered, the rate of pregnant cows obtained in winter was higher than that obtained in the summer [54]. In our study, when the OvS protocol was initiated with high P4 values, the increases in LBF and TAR values during the CP period suggests that the low conception values obtained in the aforementioned studies during the hot period are primarily due to the initiation of the OvS program with low progesterone values.

When the OvS protocol is started, corpora lutea can be found in approximately 52% of animals to be under 15 mm (D0). In this case, the subluteal function cannot sufficiently generate progesterone priming in the endometrium during the application of Ovsynch [55]. It has been determined that low P4 values due to subluteal function in the estrus cycle cause low fertility after insemination [56]. In this study, mean P4 values were determined as 0.642 ng/mL (0.10–1.42 ng/mL) in the HP and 0.85 ng/mL (0.16–1.97 ng/mL) in the CP. These low P4 values are due to the fact that when starting the OvS program, the CL is very small (<15 mm) and has a dominant follicle on the ovary with a size of approximately 2 cm. No difference was detected between the two periods (HP/CP) regarding LBF values at this P4 value concentration. However, the increase in both LBF and TAR values in the CP period as the progesterone value increased shows that when the OvS protocol was started with high P4 values, a better response was obtained than the HP. In support of the findings of the aforementioned authors, our findings also suggest that the ovaries with small corpora lutea did not receive sufficient response. Lopez-Gatius et al. [55] selectively applied OvS program (without normal selection procedures) showed significantly decreased rates of pregnancy during first AI (43% vs. 27%) and pregnancy during the second AI (36% vs. 53%) in the presence of CL. In the case of follicles, when OvS application is initiated, progesterone supplementation increases the P4 value above 2.0 ng/mL, and this value is necessary to increase the Pregnancy/AI ratio [57].

Corpus luteum blood flow was measured by Herzog et al. [7] with color Doppler during the estrus cycle, and a significant correlation was revealed between LBF and P4 concentration. The same study revealed that high luteal blood flow was vital for P4 secretion due to the close relationship between LBF and P4. However, it has also been shown that blood flow does not differ between low or high progesterone values [58]. On the other hand, in our study, the P4-2 and P4-3 group values are lower in the CP compared to HP. Honig et al. [35], using colour Doppler, showed that the dominant follicle circulation and the cycle length were affected by heat stress.

As a result, when OvS protocol is initiated with high P4 concentration, a significant difference is obtained favouring CP in terms of LBF and CL size in the comfort period (CP) compared to the summer period (HP). LBF of developing CL is higher when starting OvS with high P4 values in cows. It is thought that it may be better to apply the modified OvS programs that increase the progesterone value in cows with low progesterone values when the protocol is started. 

## 5. Conclusions

In conclusion, the degree to which luteal blood flow and size differed between cows in the CP and HP was determined by the P4 concentration at D0. However, without grouping the P4 values, there was no CP/HP difference in the LBF and the P4 values in the luteal period developing after OvS. On the 9th day after estrus, the P4 and the LBF values in the seasons CP and HP were not significantly different from each other despite the area of the CL being different. However, in cows with high P4 values (P4-2 and P4-3), higher LBF and TAR values are obtained from OvS applications in the CP compared to HP. This shows that the high P4 values were more affected by heat stress.

## Figures and Tables

**Figure 1 animals-11-02272-f001:**
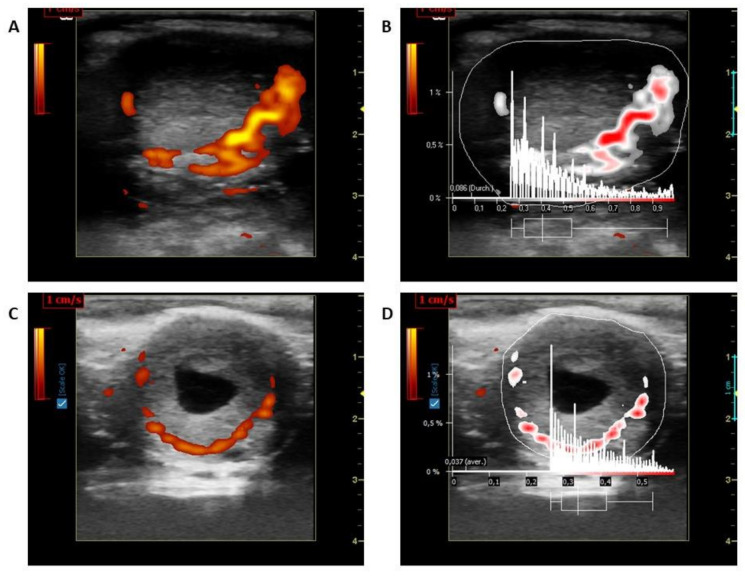
(**A**) Corpus luteum blood flow in CP; (**B**) Evaluation of Corpus luteum using Pixelflux program of Figure 1A; (**C**) Corpus luteum blood flow in HP; (**D**) Evaluation of Corpus luteum using Pixelflux program of Figure 1C.

**Table 1 animals-11-02272-t001:** Status of multiparous animals allocated to the study.

Parameter	HP (X ± SD; %)	CP (X ± SD; %)
Lactation Number	3.7 ± 1.22 (2–6)	3.6 ± 1.05 (2–6)
Days in Milk	35.0 ± 0.0 (35)	35.0 ± 0.0 (35)
Milk yield (L)	28.9 ± 3.6 (22–35)	30.4 ± 2.8 (23–35)
Milk Fat (%)	4.10 ± 0.31(3.40–4.32)	3.92 ± 0.11(3.60–4.32)
Milk Protein (%)	3.47 ± 0.01 (3.46–3.49)	3.37 ± 0.01 (3.37–3.39)
Fat/Protein Ratio	1.18	1.16
Body Condition Score (BCS)	2.9 ± 0.12 (2.75–3.00)	2.8 ± 0.12 (2.75–3.00)
Retentio secundinarium	0%	0%
Dominant Follicule + Corpus luteum < 15 mm	15/40 (37%)	13/40 (32%)
Corpus luteum >17 mm (*n*/x; %)	25/40 (62.5%)	27/40 (67.5%)
Vaginal discharge (Speculum)	0.0%	0.0%
Asymmetry Uterus * (*n*/x; %)	5/40 (12.5%)	7/40 (17%)

* Asymmetry seen found non-significant between groups (left II/right III and left II-III/right II-IV finger size *p* > 0.05; Chi-square test).

**Table 2 animals-11-02272-t002:** Study design (OvS protocol and LBF measurements).

CP & HP					
0th Day (OvSd0)	7th Day	9th Day (OeG)	10th Day	16th Day (OvSd7)	18th Day (OvSd9)
Genital examination + Blood sampling + GnRH application	PGF2α application	Blood sample + GnRH application	Follicle measurement	Doppler USG examination	Doppler USG examination + Blood sampling

OvSd0—Start of the Ovsynch program (first application); OeG—Estrus; OvSd7—7th day after diagnosed estrus; OvSd9—9th day after diagnosed estrus.

**Table 3 animals-11-02272-t003:** LBF values (mm^2^) of corpora lutea in days 7 (OvSd7) and 9 (OvSd9) regarding Hot (HP) and Comfort (CP) periods.

7th Day after OVS (OvSd7)	LBF (X ± SD)	9th Day after OVS (OvSd9)	LBF (X ± SD)
OvSd7CP (*n* = 374 *)	0.656 ± 0.295 (0.01–1.34)	OvSd9CP (*n* = 378 *)	0.691 ± 0.300 (0.06–1.71)
OvSd7HP (*n*= 383 *)	0.659 ± 0.294 (0.08–1.84)	OvSd9HP (*n* = 383 *)	0.743 ± 0.367 (0.13–1.89)
*p*	*p* > 0.05		*p* > 0.05

* Number of measurements.

**Table 4 animals-11-02272-t004:** P4 and IGF-1 results in day 9 (9 days after the end of the OvS Protocol: OvSd9) regarding Hot (HP) and Comfort (CP) periods.

Period/*n*	P_4_ (ng/mL) (X ± SD)	Period/*n*	IGF-1 (ng/mL) (X ± SD)
CP/*n* = 38	5.099 ± 2.864 (1.36–14.79)	CP/*n* = 38	6.981 ± 3.043 (3.54–14.62)
HP/*n* = 40	4.818 ± 1.904 (2.47–10.22)	HP /*n* = 40	7.274 ± 3.278 (3.00–15.07)
*p*	*p* > 0.05	*p*	*p* > 0.05

**Table 5 animals-11-02272-t005:** TAR (cm^2^) values in days 7 (OvSd7) and 9 (OvSd9) regarding Hot (HP) and Comfort (CP) periods.

7th Day after OvS	TAR (cm^2^) (X ± SD)	9th Day after OvS	Total Area (cm^2^) (X ± SD)
OvSd7CP (*n* = 374 *)	5.282 ± 1.561 (1.61–11.28)	OvSd9CP (*n* = 378 *)	5.272 ± 1.504 (2.87–10.98)
OvSd7HP (*n* = 383 *)	4.070 ± 1.303 (1.69–8.10)	OvSd9HP (*n* = 383 *)	4.526 ± 1.416 (1.22–9.04)
*p*	*p* < 0.0001	*p*	*p* < 0.0001

* Number of measurements.

**Table 6 animals-11-02272-t006:** Follicle size development and serum IGF-1 values 24 h after the last GnRH application (OeG) regarding Hot (HP) and Comfort (CP) periods.

Period	*n*	Follicule (cm^2^) (X ± SD)	*n*	IGF-1 (ng/mL) (X ± SD)
CP	30	1.674 ± 0.635 (0.75–2.55)	39	7.242 ± 3.315 (3.00–15.07)
HP	34	1.524 ± 0.325 (0.92–2.18)	39	5.881 ± 3.527 (3.27–18.85)
*p*	-	*p* > 0.05		*p* < 0.01

**Table 7 animals-11-02272-t007:** P4 values in day 0 (OvsD0) OvS protocol and LBF Arot values corresponding to P4 values (P4-1, P4-2, P4-3) on day 9 after OvS (OvSd9) and difference between HP and CP.

P_4_ (ng/mL) D0 (X ± SD)	Periods	LBF ** (mm^2^) (D18) (X ± SD)	Total Area (cm^2^) (D18) (X ± SD)
P4-1 = 0.642 ± 0.517	HP	0.715 ± 0.040	3.640 ± 1.155 (*n* = 16)
P4-1 = 0.854 ± 0.713	CP	0.611 ± 0.041	4.479 ± 1.530 (*n* = 30)
*p* > 0.05	HP/CP	*p* > 0.05	*p* < 0.05
P4-2 = 4.362 ± 0.921	HP	0.614 ± 0.047	4.474 ± 1.447 (*n* = 118)
P4-2 = 4.380 ± 1.197	CP	0.709 ± 0.031	5.064 ± 1.095 (*n* = 110)
*p* > 0.05	HP/CP	*p* < 0.004	*p* < 0.01
P4-3 = 7.368 ± 0.836	HP	0.779 ± 0.035	4.878 ± 1.341 (*n* = 55)
P4-3 = 10.338 ± 2.333	CP	0.913 ± 0.045	6.223 ± 1.825 (*n* = 49)
*p* > 0.05	HP/CP	*p* < 0.05	*p* < 0.0001

D0: OvS protocol started, LBF **: 9 days after OvS protocol (D18).

**Table 8 animals-11-02272-t008:** Correlation coefficients (r) between values measured in the comfort period (CP).

Correlation	P4-CP	LBF-CP	TAR-CP (cm^3^)
P4-CP	-	0.999; *p* < 0.05	0.999; *p* < 0.01
LBF-CP	0.999; *p* < 0.05	-	1.00; *p* < 0.001
TAR-CP	0.999; *p* < 0.01	1.00; *p* < 0.001	-

## Data Availability

The data presented in this study are available on request from the corresponding author.

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
