# Peer review of "The Effect of Comfort- and Hot-Period on the Blood Flow of Corpus Luteum (CL) in Cows Treated by an OvSynch Protocol"

_animals, 2021, doi:10.3390/ani11082272_

Round 1

Reviewer 1 Report

The authors are commended on a well-written report of what appears to be a very good data set collected to inform a very relevant question.

I have several significant queries about the methodology and a few comments regarding results and interpretation.  

Measurement of luteal blood flow refers: This was determined from 5 images of each CL but no systematic process of obtaining these multiple images to produce a valid and repeatable measure is described. This detail is necessary for a clear evaluation of the results. A figure and/or image depicting this would be helpful.

Also in relation to luteal measurements, how were cows with more than 1 CL treated? 

Also related to the measurement of blood flow, please clarify the nature of the scale (linear?) and the unit of measure reported in table 2.

Measurement of total area refers: The method of determining total area must be described. Table 6 states the unit of measure as cm cubed (cm3), which is a measure of volume rather than area. Please clarify.

The blood flow and total area results (tables 2 and 4) refers: Continuing on from the questions about measurement methods, it is not clear in the methods how the data was analysed. It would appear from the tables that all measures were treated as independent measures eg. n=378 independent measures at 9d. Please clarify this aspect, and explain whether the statistical analyses took these repeated measures from each animal into account. If not, the data must be analysed using methods which account for repeated measures. 

A table summarising the lactation number, days in milk, body condition score, genital examination findings (eg. lochia and uterine size/status, ovarian status) and production levels at the start of the study for the two groups of cows is required. 

262: It is not clear what 'positive effect on the follicle' this refers to.

The rationale of the interpretation of the results by P4 level categories is unclear. For example: 268-270: The meaning of the last phrase of this sentence is unclear ('...according to the P4 categories'). Delete or clarify. The following sentence (270-273) refers to 'these findings', but it is not clear what interpretation is being supported by the higher reported correlation coefficients.  

Lines 31 and 106: '≤0.85 (P4-1), 4.36-4.38 (P4-2) and ≥7.36 ng/ml (P4-3)': How were values of 0.85 - 4.36 and 4.38 - 7.36 treated?

Some attention to English grammar is required, especially in the discussion. There are instances of long sentences where the meaning is somewhat obscure: 212-216; 263-266; 266-268, 276-281. Please proof-read with care and use shorter, simple sentences with clear meanings.

286-288: It is not clear how the studies of varying conception rates in hot and cooler seasons 'explain the low values', or which 'values' are referred to.

292-294: The logic behind the conclusion that the blood flow difference is caused by heat stress should be explained more clearly. 

297: Replace 'which' with 'it was'

306-307: Sentence requires rephrasing, for example: 'In conclusion, the degree to which luteal blood flow and size differed between cows in the CP and HP was determined by the P4 concentration at D0.'

313: I do not follow this logic. Please rephrase for clarity of meaning.

The authors comment on the possibility of a modified ovsync protocol giving better results during heat stress period in lines 22-24 but do not discuss this in the discussion. The logic of this hypothesis should be explained in more detail in the discussion. 

Did the authors investigate whether any of the findings of the genital examination performed at the start of the study bore any relationship to the P4 levels such that they may have had predictive value in identifying cows more likely to respond well to the treatment? If so this would add value to the report.

There is an overuse of unfamiliar acronyms in this manuscript, which requires an unnecessary level of focused time and concentration to master before the content can be understood.  Please reduce the use of acronyms to a minimum. 

Reviewer 2 Report

General comments:

The present study aimed to evaluate the effect of the heat stress on some parameters related with the CL, including Doppler US. In my opinion the study is well designed, even a univariable analysis is performed. The conclusions are supported by the results. Nevertheless a few major points need be addressed to improve the final quality of the study: 1) A general improvement of the presentation is required; authors need be use only a timeline in the text and times to avoid confusion; 2) the question of the P4-1 group need be presented and discussed more deeply, once according the results, it seems that the P4 status before to apply the OvS protocol can be a relevant issue to justify the results. Please see the specific considerations. The summary/abstract should incorporate the changes.

Specific comments:

L57-58: this re-write this sentence or add another to better insert this information in the context of the introduction.

L79-94- There are significant differences between both periods for the reported values?

L106- why you consider ≤0,85ng/ml (P4-1) as threshold and not <4,36ng/ml? The P4-1 threshold is very near to 0,5ng/ml, the threshold to consider the absence of an active CL. How many cows presented P4<0,5ng/ml? this is very important, once at this postpartum stage, a significant percentage of cows can be under postpartum anestrus and other under “subluteal P4 levels). Pease use points and not comma in decimals. Check whole manuscript.

L136: After OeG? Please take care to considered 7 and 9 after  the OvS protocol (OvSd=) and after OeG. I suggest to use only one scale to avoid confusion.

L138- Once the differences were not significant, the table 2 can be removed and the overall mean of  BF for each day (7th and 9th) can be reported in text.

L143: The same for table 3

L149: The last line of the table 4 can be removed and a vs b can be added as legend to report significance.

L155 (Table 5): I don’t understand when the this IGF-1 value was taken. according to L157 it was obtained in the day of follicles measurement. It was not reported in table 1.  

L167-172 (Fi.g 1 and Table 6): the Table 6 is a repetition of fig 1 for BF and total area. Also the Figure 1 is confusing. I suggested to remove it and add a table similar to table 6 for all values or make some Graphics. This values are the core of the findings, and should be full discussed.

L180 (table 7). Please insert a legend.

L189-196: This paragraph can be removed. Why you need to describe the basis of the OvSynch program?

L209-209: Yes, it is right. We call this as the subluteal P4 levels after resumption of the postpartum ovulation; but at 35 days a proportion of the cows can still in anoestrus. You need to give the proportion of cows with P4< 0.05 ng/ml  in results and discuss this point here.

L225: “There is no publication, at our knowledge,…”

L230-231: “… when compared between the HP and CP.” can be removed.

L241: Please use here “in our study”.

L247: Two or three days?

L273-274: According to your univariable analysis, yes. But if you compare the 3 P4 groups within in the cold season (was not compared in your study) I think that you can found similar findings. In fact, the presence of a active CL and absence of subluteal P4 levels in the previous cycle can justify theses differences. Due to this occurrence external P4 is added in some OvSynch protocols

Reviewer 3 Report

Specific comments:

L29-30: language improvement needed to be comprehensive

L32-33: language improvement needed to be comprehensive. OeG day 9 needs clarification to be comprehensive.

L46: influence

L78, “2.1 Animals”. Heat stress is not only because of temperature. To provide only the mean temperature and the month of the year is not decisive. Surely, mean temperature is a basic parameter, but heat stress is applicable on animals. More evidence could be offered like mean humidity and temperature on farm, presence or absence of fans could really influence micro-climate conditions, changed in animal behavior like drop in food consumption to prove the existence of heat stress in studied animals.

L97: 35th day post partum

L100: 9th day after oestrus is named here (OeGd9), but elsewhere (table 1 and text, L102) as(OvSd9). It is confusing and needs uniform description.

L105-106: Are these P4 concentration categories pre-determined? Did these 3 categories arise from the P4 results in blood? In that case there were not values e.g. between0.85 and 4,36 ng/ml?

Table 1. While it is depicted clearly it is confusing the way initials are written. 9th day AFTER finish of OvS is named OvSd9 but is actually the 18th from the beginning of the protocol, which is named OvSd0. It should be renamed here and in text, since it is really confusing

L113-114: “total corpus luteum (CL) area (TAR)” measurements should be clearly mentioned here

Table 2 Blood flow values, units?

L140 and table 3: why only the results of day nine after OvS protocol are presented?

L161-164: This sentence is not comprehensive and again it is not clear how this classification did arise. Are the values the mean values of groups of cows? Did authors classify the P4 results in CP and HP periods in predetermined classes? Then how many cows were classified in these classes and how are these connected to BF measurements in 9th day after OvS completion day? Methodology and results presented in figure 1 and table 6 are not clear. Authors make conclusions based on this P4 classifications on Day 0 but clarification is needed.

L209: sentence is not comprehensive

L242: sentence is not comprehensive

L246-248Q this is not clear. P4 concentrations on day 7 of OvS are not presented. What this P4< 1ng/ml mean?

L254-255: previous discussion does not support this conclusion, based on presented results

L263-265: This sentence is not comprehensive

L267-268: it it not comprehensive; luteal blood flow gives different response to what?

L281-283: This sentence is not comprehensive

L286-287: previous discussion does not support this claim.

L293-294: previous discussion does not support this claim, considering the inadequate description of heat stress in this study

L294-297: This sentence is not comprehensive

L309: On 7th day after oestrus, according to table one there were no blood samples, so no P4 values could be obtained

L310-312: This sentence needs clarification or should be re-written

L313: This conclusion is not justified by the results.

Round 2

Reviewer 1 Report

The authors have responded to concerns regarding the scientific aspects of the study adequately. I therefore believe the substance of the manuscript is much improved. My main remaining critique is the English grammar and clarity of writing. I highlighted many sentences and phrases which require further review to improve clarity of meaning. 

Author Response

Response to Reviewer Comments

Reviewer 1

The authors have responded to concerns regarding the scientific aspects of the study adequately. I therefore believe the substance of the manuscript is much improved. My main remaining critique is the English grammar and clarity of writing. I highlighted many sentences and phrases which require further review to improve clarity of meaning.

Our Response: We would like to thank you very much for the reviewer to consider our paper. We carefully studied your addressed issues. We revised all issues mention at the attached pdf and you can follow our responses at your attached pdf too. Besides, Assoc. Prof. Wayne Fuller, a colleague of us and a native speaker, has checked and edited the language of manuscript.

Reviewer 2 Report

General comments
The V1 version was greatly improved at the scientific level and be able to be published in animals. Nevertheless,  several minor changes should be made mainly due to scientific writing and format before recommending its acceptance. All detected issues are reported in the specific comments section.

Specific comments
L33: “In the P4-1, but there was no difference” please re-write this sentence.
L102: Correct 1..1.18
L109 and table 1(L107-109): “in non-significant multiparous animals”. You mean “in multiparous animals” what is non-significant? The abbreviatures should be defined in the legend.  Apparently non-significant differences were observed between the two periods. I think that you need to test these differences and report it in the legend (as well the tests used) if not significant. Is ±SD or ±SEM?
L116- 117: “…after the blood was drawn.” What did this mean? I think that “the blood samples were collected at day… and day... is more clear; and (maybe) a second sentence for GnRH administration.
L120-122: Please use points for decimals instead of the comma.
L165. The Figure 1 should be reported in the text (subsection 2.3)
L189, 195, and 201: I think that the title of these tables should be re-written. E.g. (for Table 3),  Blood Flow (LBF) Values (mm2) of corpora lutea in days 7 and 9 regarding Summer (HP) and Comfort (CP) periods. Why not use P<0.001 instead of P <0.0001?
Note: corpora lutea is the plural of corpus luteum (check the whole manuscript). in the title only use the abbreviatures used in the table (or in the alternative as legend). Also revise title of Table 6 and Table 7.
L208:P<0.01; please use the format of Animals to report P values. Check whole manuscript (Text and tables)
L221:_ I suggest adding P >0.05 in the column of P4 (similar to the others columns) to avoid confusion (and remove “*”).
L231: What is P4PC, LBFCP, TAR. Please define them in the legend. Note that a table/figure should be read and interpreted independently. In the title: “Correlation coefficient (r) …”
L272: I suggest “In our study,…” instead of “In this study,…”
L280: TAR was used before. Please define the abbreviature at the first use and only use  it after this definition in whole text. Please check all abbreviatures in whole the text.  

L297-299: “No statistically significant difference was found in our study between the follicle diameters…” - “In our study, no statistically significant difference of follicle diameters was found between CP (1.67 cm) and HP (1.52 cm) …”
L322-323: I suggest also to report the correlations for LBF (r=0.xx) and TAR (r=0.yy)
L367: “…our findings also suggest that…” instead of “…our findings also reveal that…”. 
L397: “…it was revealed that LBF values of high P4 values due to heat stress…” it is not a revelation is an observation, and the values are associated to the periods (an association is not always a cause-effect) .  I suggest “…it was observed  that LBF values of the P4-2 and P4-3 group is lower in the CP than the HP.
L389: “This shows that the high P4 values were more affected by heat stress.” or “This suggest that the high P4 values are more affected by heat stress.”

Author Response

Response to Reviewer Comments

Reviewer 2

General comment

The V1 version was greatly improved at the scientific level and be able to be published in animals. Nevertheless, several minor changes should be made mainly due to scientific writing and format before recommending its acceptance. All detected issues are reported in the specific comments section.

Our Response: We are so grateful to the reviewer to let us improve our manuscript.

Specific comments:

L33: “In the P4-1, but there was no difference” please re-write this sentence.

Our Response: It was re-written.

L102: Correct 1..1.18

Our Response: It was corrected.

L109 and table 1(L107-109): “in non-significant multiparous animals”. You mean “in multiparous animals” what is non-significant? The abbreviatures should be defined in the legend.  Apparently non-significant differences were observed between the two periods. I think that you need to test these differences and report it in the legend (as well the tests used) if not significant. Is ±SD or ±SEM?

Our Response: Thanks to the reviewer. We addressed your critiques. The table, including the title and legend were revised. We clarified abbreviations by removing them and writing exact names. We tested the groups and made the required explanation at the legend. The values are expressed as X±SD.

L116- 117: “…after the blood was drawn.” What did this mean? I think that “the blood samples were collected at day… and day... is more clear; and (maybe) a second sentence for GnRH administration.

Our Response: The sentence was revised as follows “…..During the normal OvS protocol process, the blood samples were collected (Vena jugularis) into serum separator tubes (SST) from each cow at day 0 (before the start of the application: OvSd0), at day 9 (oestrous: OeG) and day 18 (9th day after OeG: OvSd9). The first GnRH application was made after the blood was collected at day 0 (OvSd0)….”

L120-122: Please use points for decimals instead of the comma.

Our Response: Thanks for your attention. We checked and made required corrections.

L165. The Figure 1 should be reported in the text (subsection 2.3)

Our Response: Thank you very much. We reported Figure 1 in sub-section 2.3.

L189, 195, and 201: I think that the title of these tables should be re-written. E.g. (for Table 3),  Blood Flow (LBF) Values (mm2) of corpora lutea in days 7 and 9 regarding Summer (HP) and Comfort (CP) periods. Why not use P<0.001 instead of P <0.0001?

Our Response: We appreciate the reviewer for the suggestion. We revised the table titles. We calculated significance as P<0.0001, and if it remains, we believe it will underline the significant level strongly.

Note: corpora lutea is the plural of corpus luteum (check the whole manuscript). in the title only use the abbreviatures used in the table (or in the alternative as legend). Also revise title of Table 6 and Table 7.

Our Response: Thanks for your suggestions. We check the addressed issues and took necessary actions.

L208:P<0.01; please use the format of Animals to report P values. Check whole manuscript (Text and tables)

Our Response: We revised the reporting style of P values according to Animals.

L221:_ I suggest adding P >0.05 in the column of P4 (similar to the others columns) to avoid confusion (and remove “*”).

Our Response: Thank you. We did your suggestion and removed “*”.

L231: What is P4PC, LBFCP, TAR. Please define them in the legend. Note that a table/figure should be read and interpreted independently. In the title: “Correlation coefficient (r) …”

Our Response: We revised to clarify it. Abbreviations of CP, TAR, LBF and P4 were explained in the text.

L272: I suggest “In our study,…” instead of “In this study,…”

Our Response: Thanks. We revised it.

L280: TAR was used before. Please define the abbreviature at the first use and only use  it after this definition in whole text. Please check all abbreviatures in whole the text. 

Our Response: We checked all abbreviatures in the whole text and revised the required points.

L297-299: “No statistically significant difference was found in our study between the follicle diameters…” - “In our study, no statistically significant difference of follicle diameters was found between CP (1.67 cm) and HP (1.52 cm) …”

Our Response: Thanks very much. We revised it according to your suggestion.

L322-323: I suggest also to report the correlations for LBF (r=0.xx) and TAR (r=0.yy)

Our Response: We did your advice, thanks.

L367: “…our findings also suggest that…” instead of “…our findings also reveal that…”.

Our Response: We revised it.

L397: “…it was revealed that LBF values of high P4 values due to heat stress…” it is not a revelation is an observation, and the values are associated to the periods (an association is not always a cause-effect) .  I suggest “…it was observed that LBF values of the P4-2 and P4-3 group is lower in the CP than the HP.

Our Response: Thank you. We revise it.

L389: “This shows that the high P4 values were more affected by heat stress.” or “This suggest that the high P4 values are more affected by heat stress.”

Our Response: We revised it as “were”.

Reviewer 3 Report

Authors responded to comments and suggestions and made considerable changed in the whole manuscript.

Additions improved the manuscript, but they look like a posteriori additions, which in many parts confuse more than enlight the presented research

e.g.

L72-73: this sentence is attaches without connection to previous text and sets a different classification option (winter-summer) than the proposed earlier (comfort-heat period). It is confusing

L119: it should be clear that this P4 classification derived from the P4 measurements of the first day blood sampling.

L29 blood samples were collected and during oestrus (OeG)…

but in L117-118: “Then blood was drawn again on the 9th day (OvSd9) after oestrous (D18)”

and in table 2  (L132) 3 sampling times are presented!

Different blood sampling times are presented in the text, which makes it difficult to understand the plan of the authors

Overall, language is improved, but there are still mistakes (e.g. L33, L355-56,…)

Author Response

Response to Reviewer Comments

Reviewer 3

Authors responded to comments and suggestions and made considerable changed in the whole manuscript.

Additions improved the manuscript, but they look like a posteriori additions, which in many parts confuse more than enlight the presented research

Our Response: We would like to convey our sincere thanks to the reviewer for providing us with a chance to improve our manuscript.

e.g.

L72-73: this sentence is attaches without connection to previous text and sets a different classification option (winter-summer) than the proposed earlier (comfort-heat period). It is confusing

Our Response: We corrected it as “….OVS protocols in comfort and heat periods.”.

L119: it should be clear that this P4 classification derived from the P4 measurements of the first day blood sampling.

Our Response: It was clarified as follows: The P4 (ng/ml) mean values at day 0 (OvSd0) were classified as f…….

L29 blood samples were collected and during oestrus (OeG)…

Our Response: It was clarified.

but in L117-118: “Then blood was drawn again on the 9th day (OvSd9) after oestrous (D18)”

Our Response: It was revised.

and in table 2  (L132) 3 sampling times are presented!

Our Response: Sampling times were clarified in text.

Different blood sampling times are presented in the text, which makes it difficult to understand the plan of the authors

Our Response: It was clarified.

Overall, language is improved, but there are still mistakes (e.g. L33, L355-56,…)

Our Response: Thanks. Assoc. Prof. Dr. Wayne FULLER, a colleague of ours and a native speaker, has checked and edited the manuscript.